# Beyond Univariate Calibration: Verifying Spatial Structure in Ensembles of Forecast Fields

Joshuah Jacobson[1], William Kleiber[1], Michael Scheuerer[2,3], and Joseph Bellier[2,3]

[1]Department of Applied Mathematics, University of Colorado Boulder
[2]Cooperative Institute for Research in Environmental Sciences, University of Colorado Boulder
[3]Physical Sciences Laboratory, National Oceanic and Atmospheric Administration, Boulder, Colorado

**Correspondence:** Joshuah Jacobson (Josh.Jacobson@Colorado.edu)

**Abstract.** Most available verification metrics for ensemble forecasts focus on univariate quantities. That is, they assess whether the ensemble provides an adequate representation of the forecast uncertainty about the quantity of interest at a particular location and time. For spatially-indexed ensemble forecasts, however, it is also important that forecast fields reproduce the spatial structure of the observed field, and represent the uncertainty about spatial properties such as the size of the area for which heavy precipitation, high winds, critical fire weather conditions, etc. are expected. In this article we study the properties of the fraction of threshold exceedance (FTE) histogram, a new diagnostic tool designed for spatially-indexed ensemble forecast fields. Defined as the fraction of grid points where a prescribed threshold is exceeded, the FTE is calculated for the verification field, and separately for each ensemble member. It yields a projection of a – possibly high-dimensional – multivariate quantity onto a univariate quantity that can be studied with standard tools like verification rank histograms. This projection is appealing since it reflects a spatial property that is intuitive and directly relevant in applications, though it is not obvious whether the FTE is sufficiently sensitive to misrepresentation of spatial structure in the ensemble. In a comprehensive simulation study we find that departures from uniformity of the FTE histograms can indeed be related to forecast ensembles with biased spatial variability, and that these histograms detect shortcomings in the spatial structure of ensemble forecast fields that are not obvious by eye. For demonstration, FTE histograms are applied in the context of spatially downscaled ensemble precipitation forecast fields from NOAA's Global Ensemble Forecast System.

*Copyright statement.* TEXT

## 1 Introduction

Ensemble prediction systems like the ECMWF ensemble (Buizza et al., 2007) or NOAA's Global Ensemble Forecast System (GEFS; Zhou et al., 2017) are now state of the art in operational meteorological forecasting at weather prediction centers worldwide. One of the goals of ensemble forecasting is the representation of uncertainty about the state of the atmosphere at a future time (Toth and Kalnay, 1993; Leutbecher and Palmer, 2008), and verification metrics are required that can assess to what extent this goal is achieved. For univariate quantities, i.e. if forecasts are studied separately for each location and each

forecast lead time, diagnostic tools like verification rank histograms (Anderson, 1996; Hamill, 2001) or reliability diagrams (Murphy and Winkler, 1977) can be used to check whether ensemble forecasts are calibrated, i.e. statistically consistent with the values that materialize.

When entire forecast fields are considered, aspects beyond univariate calibration are important. For example, ensembles that yield reliable probabilistic forecasts at each location may still over- or under-forecast regional minima/maxima if their members exhibit an inaccurate spatial structure (e.g., Feldmann et al., 2015, their Fig. 6). For weather variables like precipitation, which are used as inputs to hydrological forecast models, it is crucial that accumulations over space and time (and the associated uncertainty) are predicted accurately, and this again requires an adequate representation of spatial structure and temporal persistence of precipitation by the ensemble.

There is an added difficultly for forecasters in that misrepresentation of the spatial structure of weather variables by ensemble forecast fields may not be discernible by eye. For example, consider the simulated fields in Fig. 1: perhaps one of these forecast fields has a clearly different spatial correlation length than the verification, but we suspect that even the sharp-eyed reader cannot distinguish between the remaining fields with confidence. Even if the differences are obvious, a quantitative verification metric is required to objectively compare different forecast systems or methodologies.

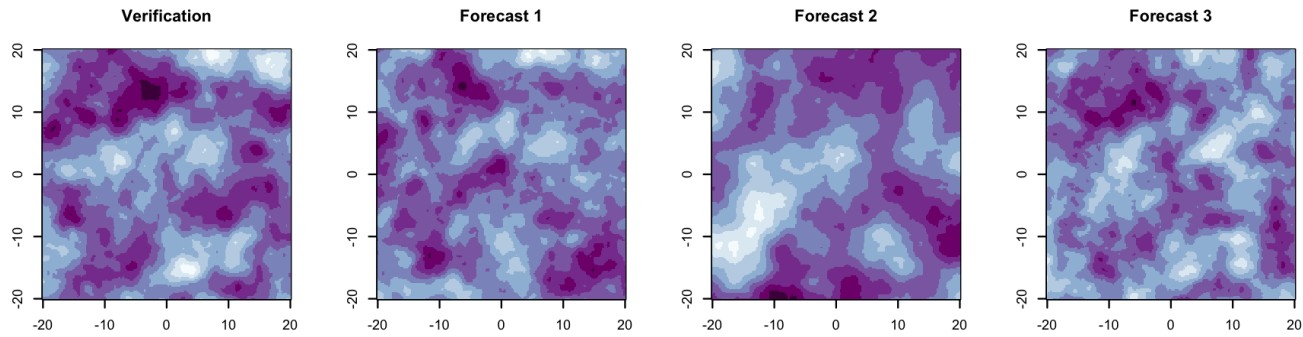

**Figure 1.** Simulated verification field and three associated forecast fields (arbitrary color scale) in which the spatial correlation length is either the same as for the verification, 10% miscalibrated, or 50% miscalibrated. Can you tell which is correct?

Several multivariate generalizations of verification rank histograms such as minimum spanning tree histograms (Smith and Hansen, 2004; Wilks, 2004), multivariate rank histograms (Gneiting et al., 2008), average-rank and band-depth rank histograms (Thorarinsdottir et al., 2016), and copula probability integral transform histograms (Ziegel and Gneiting, 2014) have been proposed and allow one to assess different aspects of multivariate calibration. They are all based on different projections of the multivariate quantity of interest onto a univariate quantity that can then be studied using standard verification rank histograms. Unfortunately, most of these projections do not allow an intuitive understanding of exactly what multivariate aspect is being assessed, and none are tailored to the special case where the multivariate quantity of interest is a spatial field. Two recent papers (Buschow et al., 2019; Buschow and Friederichs, 2020) propose a wavelet-based verification approach in which wavelet transformations of forecast and observed fields are performed to characterize and compare the fields' texture.

The authors demonstrate that this approach is able to detect differences in the spatial correlation length similar to those shown in Fig. 1. Kapp et al. (2018) define a skill score based on wavelet spectra and study the score differences between a randomly selected ensemble member and the verification field in order to detect possible deficiencies in the texture of the forecast fields. Our goal is similar, but the approach studied here follows the idea of defining a projection from the multivariate quantity (here: a spatial field) to a univariate quantity that can be analyzed via verification rank histograms. Our main focus is on the probabilistic nature of the forecasts, that is, we want to test whether the ensemble adequately represents the *uncertainty* about spatial quantities.

The projection underlying the verification metric studied here is based on threshold exceedances of the forecast and observation fields. This binarization of continuous weather variables is common in spatial forecast verification (see Gilleland et al., 2009) as it allows one to study, for example, low, intermediate, and high precipitation amounts separately. In the context of deterministic forecast verification, Roberts and Lean (2008) define the fractions skill score (FSS) based on the fraction of threshold exceedances (FTEs) within a certain neighborhood of every grid point, and use it to examine at which spatial scale the forecast FTEs become skillful. Scheuerer and Hamill (2018) use a similar concept to study whether an ensemble of forecast fields adequately represents spatial forecast uncertainty. They calculate the FTE for all ensemble members and the verifying observation field, and study verification rank histograms of the resulting univariate quantity in order to diagnose the advantages and limitations of different statistical methods to generate high-resolution ensemble precipitation forecast fields based on lower resolution NWP model output. The FTE is an interpretable quantity that is highly relevant in applications where the fraction of the forecast domain for which severe weather conditions are expected (e.g., heavy rain, extreme wind speeds, etc.) may be of interest. However, it is not obvious whether FTE histograms are sufficiently sensitive to misrepresentation of the spatial structure by the ensemble, and the goal of the present paper is to investigate this discrimination ability in detail.

In section 2, we describe the calculation of the FTE and the construction of the FTE histogram in detail. In section 3, a simulation study is designed and implemented that allows us to analyze the discrimination capability of the FTE histograms with regard to spatial structures. In section 4, we demonstrate the utility of FTE histograms in the context of spatially downscaled ensemble precipitation forecast fields from NOAA's Global Ensemble Forecast System. A discussion and concluding remarks are given in section 5.

## 2 The fraction of threshold exceedance metric

Let $Z(s)$ be a scalar field on a domain $s \in D$. Here, we describe a strategy of studying exceedances of $Z$ at various thresholds. That is, we focus interest in statistics based on $\mathbf{1}_{\{Z(s)>\tau\}}$ for a given threshold $\tau \in \mathbb{R}$. In the domain $D$, we define the fraction of threshold exceedance (FTE) as the fraction of all points at which $\tau$ is exceeded. Specifically, let

$$
\begin{aligned}
\mathrm{FTE}(Z,\tau) &= \frac{1}{|D|}\int_D \mathbf{1}_{\{Z(s)>\tau\}}(s)ds \\
&= \frac{1}{n}\sum_{j=1}^{n}\mathbf{1}_{\{Z(s)>\tau\}}(s_j)
\end{aligned}
\tag{1}
$$

where the first equality represents the idealized continuous spatial process definition, while the second reflects the discrete nature of spatial sampling in an operational probabilistic forecasting context with $D = \{s_1, \ldots, s_n\}$. The resulting univariate quantity can be evaluated by common univariate verification metrics (Scheuerer and Hamill, 2018).

Suppose we have a $k$-member ensemble $Z_1(s), \ldots, Z_k(s)$ and associated verification field $Z_0(s)$ (e.g., observation or analysis) all on $D$; let $\pi = \{\text{FTE}(Z_0, \tau), \ldots, \text{FTE}(Z_k, \tau)\}$. Note that $\pi$ depends on the threshold, but for ease of exposition we do not include this dependence in notation. We call $r$ the rank of the verification FTE relative to the set of verification and ensemble forecast FTEs, or the rank of $\text{FTE}(Z_0, \tau)$ in $\pi$. There are three cases of interest when computing $r$: (1) no ties exist in $\pi$, (2) ties exist among a subset of $\pi$ that includes $\text{FTE}(Z_0, \tau)$, or (3) there is only one unique value in $\pi$. In the first case no special action

is required, and in the second case ties in rank are simply broken uniformly at random. The third case arises when all ensemble members have the exact same FTE as the verification, as may occur, for example, when the precipitation amount reported by the verification and predicted by all ensemble members is below the threshold $\tau$ everywhere in $D$. Instances of this case are completely uninformative for the purpose of diagnosing miscalibration and can be discarded.

    Gathering ranks over $N$ instances of forecast/verification pairs, $r_1, \ldots, r_N$, a natural way to communicate the FTE rank

behavior is through a histogram (termed *FTE histogram* by Scheuerer and Hamill (2018)) over the $k+1$ possible ranks. Its construction is akin to that of the univariate verification rank histogram discussed in Anderson (1996) and Hamill (2001), but the latter only evaluates the marginal distribution of the ensemble. The FTE histogram behaves similar to the univariate verification rank histogram under marginal miscalibration in that overpopulated low (high) bins are an indication of an over-forecast (under-forecast) bias, and a ∪–shaped (∩–shaped) histogram is an indication of an under-dispersed (over-dispersed)

ensemble. However, it is also sensitive to misrepresentation of spatial correlations by the ensemble forecast fields. To see this, consider first the extreme case where the forecast fields are spatially uncorrelated (i.e., spatial white noise) while the verification fields have maximal spatial correlations. In this setup, if $\tau$ is equal to the climatological median of the marginal distributions at each grid point, the FTE for each ensemble member is close to $0.5$ while the FTE for the verification field is either 0 or 1, with equal probability. The associated FTE histogram is ∪–shaped with half of the cases in the lowest bin and the

other half in the highest bin. If $\tau$ is equal to the $95th$ climatological percentile, the FTE of each ensemble member is close to $0.05$ and the FTE of the verification field is 0 with probability 0.95 and 1 with probability 0.05. The associated FTE histogram is ∪–shaped *and* skewed, with 95% of all cases in the lowest bin and 5% of all cases in the highest bin. For $\tau$ equal to the $5th$ climatological percentile, the skewness is in the other direction with 5% (95%) of all cases in the lowest (highest) bin. In a more realistic situation, where both forecast end verification fields are spatially correlated but the spatial correlations of the

forecast fields are too weak (i.e., they exhibit too much spatial variability), we can still expect to see a somewhat ∪–shaped FTE histogram since the verification FTE values are more likely to assume extreme ranks than the ensemble FTE values. For large values of $\tau$, the lower bins will be more populated, for small values of $\tau$, the higher bins will be more populated. Conversely, if the spatial correlations of the forecast fields are too strong, the verification ranks will over-populate the central bins, slightly shifted upward or downward from the center depending on $\tau$. An ensemble that is marginally and spatially calibrated (i.e., the

strength of spatial correlations within each forecast field matches that of the verification) will result in a flat FTE histogram.

If the marginal forecast distributions are miscalibrated, the resulting effects on the rank of the verification FTE are superimposed on those caused by misrepresentation of spatial correlations. This complicates interpretation because it is often impossible to disentangle the different sources of miscalibration (this loss of information is an inevitable consequence of projecting a multivariate quantity onto a univariate one), and it can even happen that different effects cancel each other out. For example, ensemble forecast fields which are both under-dispersive and have too strong spatial correlations may result in flat FTE histograms. This serves as a reminder that – as in the univariate case – a flat histogram is a necessary but not a sufficient condition for probabilistic calibration. It simply indicates that the verification and the ensemble are indistinguishable with regard to the particular aspect of the forecast fields (here: exceedance of a prespecified threshold) assessed by this metric. Systematic over- or under-forecast biases can be accounted for by using different (depending on the respective climatology) threshold values $\tau$ for the forecast and verification fields. We are not aware of an equally straightforward way to account for dispersion errors, so we encourage users to always check the marginal forecast distributions first, and then study FTE histograms for different thresholds, possibly in conjunction with other multivariate verification metrics in order to obtain a comprehensive picture of the multivariate properties of the ensemble forecasts.

While the FTE histogram is a useful visual diagnostic tool, a quantitative measure for studying departures from uniformity is desirable. Akin to Keller and Hense (2011), we fit a beta distribution to the histogram values (transformed to the unit interval) and characterize the histogram shape based on the $\beta$-*score* and $\beta$-*bias*, respectively defined as

$$\beta_S = 1 - \sqrt{\frac{1}{a \cdot b}}, \quad \beta_B = b - a, \tag{2}$$

where $a$ and $b$ are the two distribution parameters. Since histogram values only occur at discrete points in $[0,1]$, parameter estimation methods will incur some bias due to the lack of data on the interior of adjacent ranks. Thus, we stochastically disaggregate the (transformed) ranks $r_1, \ldots, r_N$ to continuous values in $[0,1]$ (see Appendix A for details) and fit a beta distribution via maximum likelihood. Together, the $\beta$-score and $\beta$-bias provide a pair of succinct descriptive statistics which communicate the visual characteristics of the histogram, and therefore the ensemble's calibration properties. In the ideal case, $\beta_S$ and $\beta_B$ are both exactly zero, indicating that the FTE histogram is perfectly uniform. In practice, these metrics are never exactly zero. The resulting set of possible deviations and broad interpretations of the corresponding histogram shapes are outlined in Table 1. With the $\beta$-score and $\beta$-bias, we have an easily interpreted measure of spatial forecast calibration.

In summary, the FTE metric is composed of three steps: (1) calculate the FTE of each verification and ensemble forecast field, (2) construct an FTE histogram over available instances of forecast and verification times, and (3) derive the $\beta$-score and $\beta$-bias from the stochastically disaggregated FTE histogram to characterize departure from uniformity.

## 3   Simulation study

In this section we consider an extensive simulation study to assess the ability of the proposed FTE histogram to diagnose deficiencies in the representation of spatial variability by the ensemble forecast fields. Our simulations will be based on multivariate Gaussian processes where the notion of "spatial variability" can be quantified in terms of a correlation length parameter. The

**Table 1.** Characterization of FTE histogram shapes via $\beta$-score and $\beta$-bias and their interpretation with regard to potential deficiencies of the ensemble forecast fields.

| Histogram | Parameters | Score & Bias | Interpretation |
|---|---|---|---|
| Uniform | $a = b = 1$ | $\beta_S = \beta_B = 0$ | Ensemble FTEs consistent with verification FTE |
| $\cup$–shaped | $a, b < 1$ | $\beta_S < 0$ | Under-dispersed marginal distributions OR excessive spatial variability |
| $\cap$–shaped | $a, b > 1$ | $\beta_S > 0$ | Over-dispersed marginal distributions OR insufficient spatial variability |
| Right-skewed | $a < b$ | $\beta_B > 0$ | Over-forecast bias OR excessive spatial variability at high thresholds |
| Left-skewed | $a > b$ | $\beta_B < 0$ | Under-forecast bias OR insufficient spatial variability at high thresholds |

Skewness is exaggerated by high thresholds; see text for more detail.

various meteorological quantities of interest such as precipitation and wind speeds can be quite heterogeneous and spatially nonstationary over the study domain. However, since we study the spatial structure of threshold exceedances, a suitable choice of thresholds can mitigate these effects to a degree that multivariate, stationary Gaussian processes can be viewed as a sufficiently flexible model for simulating realistic spatial fields. To see this, consider a strictly positive and continuous variable $Z(s)$ at two spatial locations $s = s_1, s_2$ with possibly unequal continuous cumulative distribution functions $F_1$ and $F_2$, respectively. Rather than considering a spatially-constant threshold such as 10 m/s for wind gusts, we can use a location-dependent threshold, say the 90% climatological quantiles $q(s_1)$ and $q(s_2)$ representing local characteristics. Then both quantities $\mathbf{1}_{\{Z(s_i) > q(s_i)\}}$, $i = 1, 2$, are identically distributed Bernoulli(0.1) random variables. Exploiting a standard Gaussian probability integral transformation method, we note that $\Phi^{-1}(F(Z(s_i)))$ is a standard normal random variable, where $\Phi$ is the cumulative distribution function of a standard normal. Thus, the original probability of threshold exceedance can be written

$$P(Z(s_i) > q(s_i)) = P(F(Z(s_i)) > F(q(s_i))) = P(\Phi^{-1}(F(Z(s_i))) > \Phi^{-1}(F(q(s_i))) = P(X > \Phi^{-1}(0.9)) \tag{3}$$

where $X$ is a standard normal. Thus, we have shown that a field of random variables with continuous, possibly distinct local probability distributions can be transformed to standard Gaussian marginal distributions, and using local quantiles as the threshold is then equivalent to a spatially-constant threshold on the transformed variables. For weather variables with discrete-continuous marginal distributions (e.g., precipitation), this direction of the transformation is not quite as straightforward. Conversely, however, simulated fields from Gaussian processes can always be transformed to any desired marginal distributions (including discrete-continuous ones). In our ensuing simulations studies we therefore consider stationary spatial Gaussian processes as representing forecast and verification fields.

The main technical difficulty in setting up the simulation study is in generating multiple, stationary Gaussian random fields that have different correlation lengths while being correlated with each other. That is, we would like to generate $Z_0(s)$ and $Z_1(s)$ in such a way that $\text{Cov}(Z_0(s), Z_1(s)) > 0$ (representing that the forecast field is correlated with the verification field) and where $Z_0$ and $Z_1$ have possibly distinct correlation lengths (representing that the forecast field is spatially miscalibrated). A natural approach is to use multivariate random field models.

### 3.1 Multivariate Gaussian processes

We call a vector of processes $(Z_0(s), Z_1(s), \ldots, Z_k(s))$ a multivariate Gaussian process if its finite-dimensional distributions are multivariate normal. We focus on second-order stationary mean zero multivariate Gaussian processes in that $E(Z_i(s)) = 0$ for all $i = 0, \ldots, k$ and $s \in D$. Stationarity implies that the stochastic process is characterized by

$$C_{ij}(h) = \text{Cov}(Z_i(s+h), Z_j(s)), \quad \text{for all } h \text{ such that } s+h \in D, \tag{4}$$

which are called covariance functions for $i = j$ and cross-covariance functions for $i \neq j$. Not all choices of functions $C_{ij}$ will result in a valid model, in particular we require that the matrix of functions $\mathbf{C}(h) = (C_{ij}(h))_{i,j=0}^k$ be a nonnegative definite matrix function, the technical definition of which can be found in Genton and Kleiber (2015).

There are many models for multivariate processes (Genton and Kleiber, 2015), and here we exploit a particular class called the multivariate Matérn (Gneiting et al., 2010; Apanasovich et al., 2012). We rely on the popular Matérn correlation function

$$M(d|\nu, a) = \frac{2^{1-\nu}}{\Gamma(\nu)} \left(\frac{d}{a}\right)^\nu K_\nu \left(\frac{d}{a}\right) \tag{5}$$

where $\Gamma$ is the gamma function, $K_\nu$ is the modified Bessel function of the second kind of order $\nu$ and $d$ is a non-negative scalar. Parameters have interpretations as a smoothness ($\nu$), and spatial range or correlation length ($a$). The multivariate Matérn correlation function is defined as

$$C_{ii}(h) = \sigma_i^2 M(\|h\| | \nu_i, a_i), \quad \text{for } i = 0, \ldots, k, \tag{6}$$

and

$$C_{ij}(h) = C_{ji}(h) = \rho_{ij} \sigma_i \sigma_j M(\|h\| | \nu_{ij}, a_{ij}), \quad \text{for } 0 \leq i \neq j \leq k \tag{7}$$

where $\| \cdot \|$ is the Euclidean norm. In this latter equation, $\rho_{ij} \in [-1, 1]$ is the co-located cross-correlation coefficient. Interpretation of the cross-covariance parameters requires spectral techniques (Kleiber, 2017).

### 3.2 Simulation setup

Simultaneously simulating the verification field $Z_0(s)$ and all forecast fields $Z_1(s), \ldots, Z_k(s)$ is difficult due to the high-dimensional joint covariance matrix. Instead, we approach simulations by jointly simulating the verification fields $Z_0(s)$ and the (scaled) ensemble mean field, $Z_M(s)$ from a bivariate Matérn model. We then perturb the mean field with independent univariate Gaussian random fields to generate an 11-member ensemble, $k = 11$.

The simulation setup follows a series of steps:

1. Generate $Z_0$ and $Z_M$, the verification and (scaled) ensemble mean as a mean zero bivariate Gaussian random field with multivariate Matérn correlation length parameters $a_0$, $a_M$, and $a_{0M} = \sqrt{a_0 a_M}$, smoothness parameters $\nu_0 = \nu_{0M} = \nu_M = 1.5$, and co-located correlation coefficient $\rho_{0M} = \omega = 0.8$.

2. Generate 11 independent mean zero Gaussian random fields $W_1(s), \ldots, W_{11}(s)$ with Matérn covariance having correlation length $a = a_M$ and smoothness $\nu = \nu_M = 1.5$.

3. The ensemble member fields $Z_1(s), \ldots, Z_{11}(s)$ are constructed as

$$Z_i(s) = \omega Z_M(s) + \sqrt{1-\omega^2} W_i(s), \quad i = 1, \ldots, 11. \tag{8}$$

The third step implies that each field in the ensemble is a Gaussian process with mean zero, variance one, correlation length $a_M$, smoothness $\nu_M$, and univariate "forecast skill" controlled by the parameter $\omega$ (see Appendix B). Note that by choosing the co-located correlation coefficient $\rho_{0M} = \omega$, the correlation between the verification and each ensemble member is $\omega^2$, the same as the correlation between ensemble members themselves. That is, $\text{Cov}[Z_i, Z_j] = \omega^2$ for $i, j = 0 \ldots, 11$ when $i \neq j$ (derivation in Appendix C), and thus the ensemble forecasts are calibrated in the univariate sense.

In this study, fields were constructed on a square grid over the domain $[-20, 20] \times [-20, 20]$ with resolution 0.2. Verification-ensemble samples were collected by repeating the simulation above 5000 times for each combination of

$$a_0 \in \{1, 1.5, \ldots, 3.5, 4\}, \quad a_M \in \{0.5a_0, 0.6a_0, \ldots, 1.4a_0, 1.5a_0\},$$

resulting in a total of 77 experiments. Note that in practice, each sample corresponds to a date for which forecasts have been issued and verifying observations are available, meaning the sample size is governed by the time period for which the verification is performed. For each experiment, FTE histograms were constructed from the 5000 samples using each of $\tau \in \{0, 0.5, \ldots, 3.5, 4\}$. That is, for a given $a_0$ and $a_M$ we analyzed nine FTE histograms, for a total of 693 histograms across all experiments.

### 3.3 Simulation analysis

The question of primary interest in this analysis is whether the FTE histogram accurately identifies miscalibration of ensemble correlation lengths.

#### 3.3.1 Illustrative examples of FTE histograms

First, we study the discrimination ability of the FTE histogram in something of an exaggerated setting, where the miscalibration is obvious. We choose the median of the marginal distribution as the threshold (i.e., $\tau = 0$) and a verification correlation length of 2. On this grid, binary fields produced in this way appear qualitatively similar to the binary precipitation fields analyzed later in this manuscript (see Fig. 7). The correlation length ratio is the ratio of the ensemble correlation length to that of the verification field. We study ensembles with too small of a correlation length using ratio 0.5 (Fig. 2, row A), correct correlation length using ratio 1.0 (Fig. 2, row B), and too large of a correlation length using ratio 1.5 (Fig. 2, row C). Corresponding FTE histograms are then constructed with respect to these three ratios using 5000 verification-ensemble samples in each case. This revealing example is depicted in Fig. 2 and behaves as described in Table 1, where the FTE histogram takes a $\cup$-shape ($\cap$-shape) when the ensemble correlation length is too small (large), indicating excessive (insufficient) spatial variability. As

desired, the FTE histogram is approximately flat when the ensemble fields have the same correlation length as the verification field.

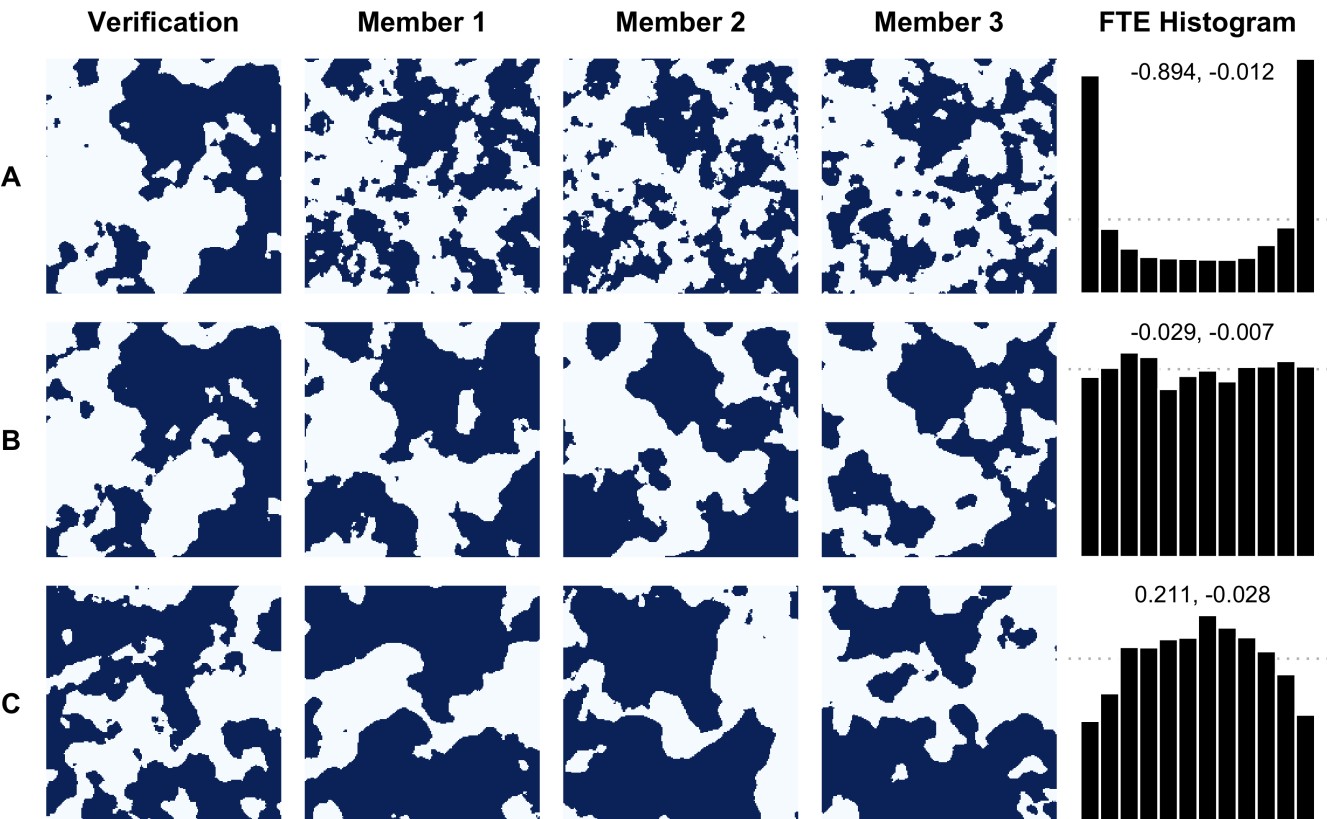

**Figure 2.** Example binary exceedance verification field and a subset of ensemble fields with representative FTE histogram for threshold $\tau = 0$ found using 5000 samples. Dark blue regions indicate threshold exceedance. All verification fields have correlation length $a_0 = 2$ and ensemble fields have correlation length $a_M = 1, 2, 3$ in rows A, B, and C respectively. FTE histograms are density histograms with dotted line $y = 1$ and corresponding $\beta$-score (left) and $\beta$-bias (right) annotated.

While the FTE histogram is able to correctly identify the obvious miscalibration of the ensemble for the scenario in Fig. 2, one could likely draw the same conclusions by visual inspection and would not use the FTE histogram for these fields in practice. However, ensemble forecast models are not generally so grossly miscalibrated; though a true correlation length ratio does not exist in reality, the theoretical ratio will often be much closer to unity. Therefore, the true utility of the FTE histogram is realized when the miscalibration is not so visually obvious. This more realistic example is illustrated in Fig. 3 where the above experiment is repeated using different correlation length ratios. In row A, the ensembles have ratio 0.9 and the resulting FTE histogram is still noticeably ∪-shaped. The ratio in row B is 1.0 which yields a flat FTE histogram. In row C, the ratio is

1.1 and the FTE histogram is noticeably ∩-shaped. Again, these results are consistent with Table 1, and we conclude that the FTE histogram maintains accurate discrimination ability even when ensemble members are only slightly miscalibrated.

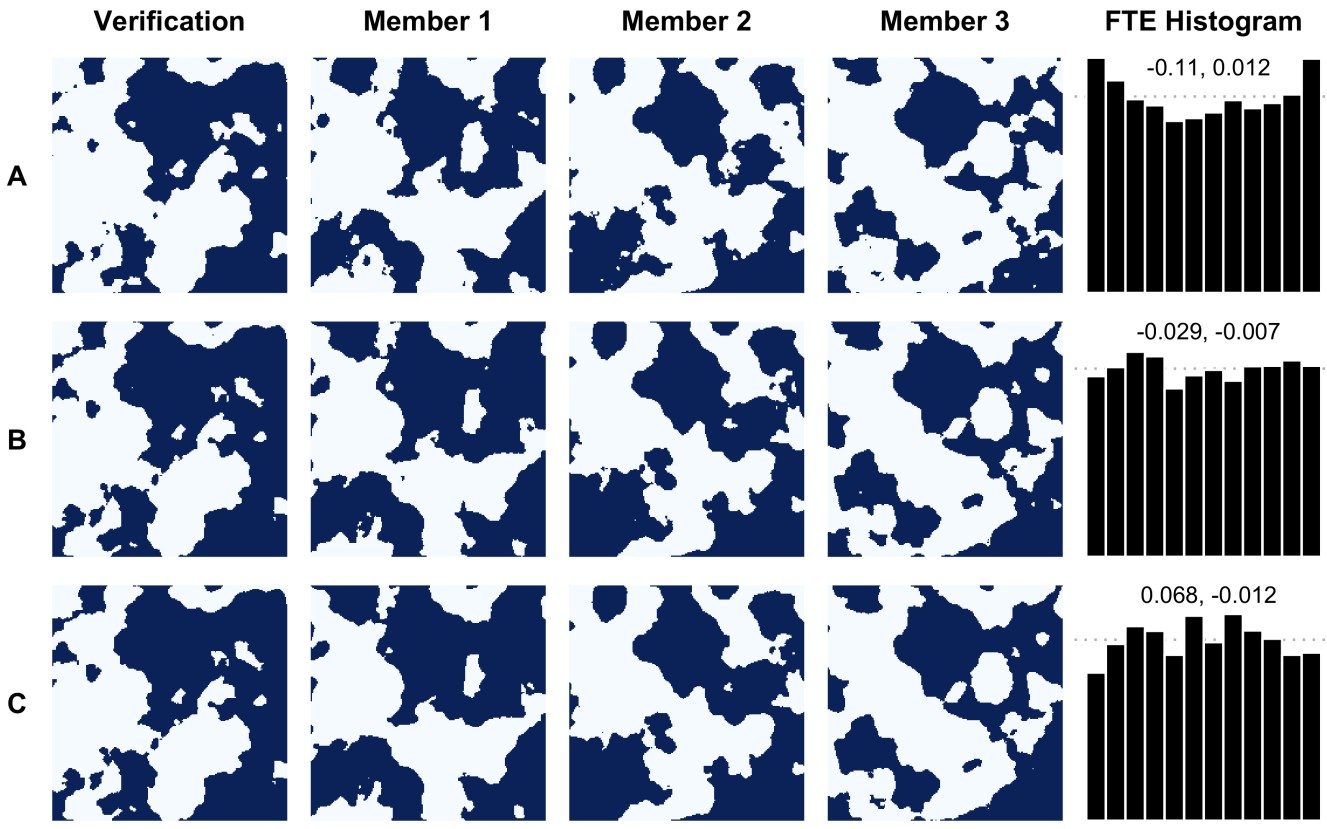

**Figure 3.** As Fig. 2, but ensemble fields have correlation length $a_M = 1.8, 2, 2.2$ in rows A, B, and C respectively.

Of course, one may often want to use a threshold parameter other than the median of the marginal distributions. The choice of $\tau$ is somewhat application specific; for example, it can be chosen such as to focus on high precipitation amounts. Thus, it is important that the FTE histogram maintains discrimination ability for different choices of $\tau$. For a visual example, the same experiment depicted in Fig. 3 is repeated in Fig. 4, but with FTE histograms constructed using $\tau = 2$ (equivalent to two standard deviations from the mean in this case). When the ensemble fields have a correlation length that is slightly too small

(row A), the resulting FTE histogram is ∪-shaped and has a slight right-skew due to the higher threshold, but correctly indicates excessive spatial variability. When the ensemble exhibits insufficient spatial variability, i.e. correlation length is slightly too large (row C), the FTE histogram is ∩-shaped and somewhat left-skewed. Reassuringly, the FTE histogram remains flat when the ensemble fields share the same correlation length as the verification fields (row B). While these results are in agreement with Table 1, the effect of the threshold can be studied more generally using the estimated $\beta$-score and $\beta$-bias.

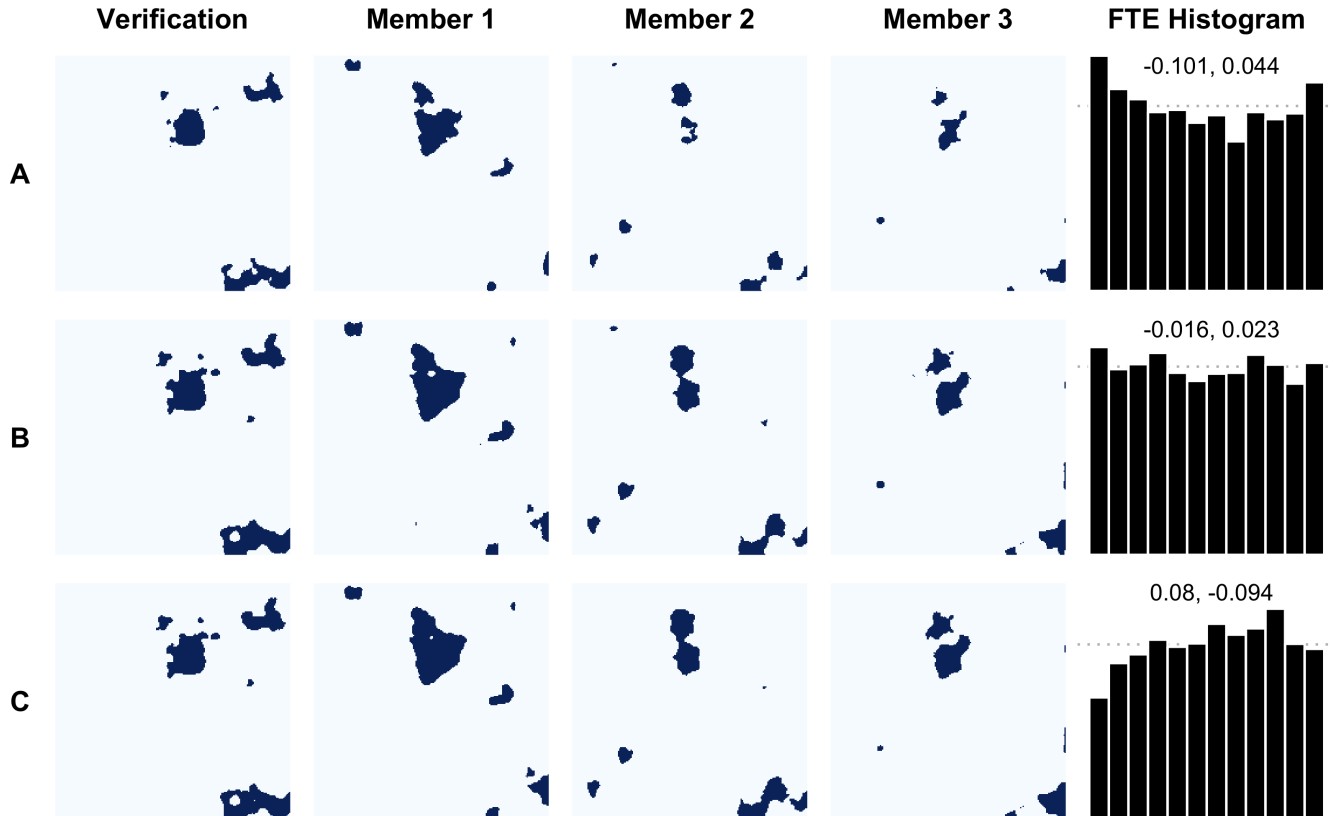

**Figure 4.** As Fig. 3, but with threshold $\tau = 2$.

### 3.3.2 Quantifying deviation from uniformity

Recall that we propose quantifying the shape of the FTE histogram with the $\beta$-score and $\beta$-bias. When $\beta_S = \beta_B = 0$, the FTE histogram is perfectly uniform. How do these metrics change as the threshold $\tau$ increases? Figure 5 demonstrates how the $\beta$-score and $\beta$-bias vary over increasing thresholds for different correlation length ratios; the estimated $\beta$-distribution parameters are also depicted for comparison with Table 1. Where provided, confidence intervals were estimated via the the nonparametric bootstrap method (see Delignette-Muller and Dutang, 2015; Cullen and Frey, 1999). When the correlation length ratio is 1.0, both the $\beta$-score and $\beta$-bias are approximately zero for every choice of $\tau$, correctly indicating a spatially calibrated ensemble. When the correlation length ratio is less (greater) than 1.0, the $\beta$-scores are themselves generally less (greater) than zero, indicating excessive (insufficient) spatial variability. As previously discussed, the $\beta$-bias becomes more pronounced at higher thresholds, thus highlighting the inextricable link between threshold and skewness. Above a very high threshold of about three standard deviations (i.e., $\tau \approx 3$), both metrics exhibit a tendency toward zero. This is partially due to the fact that the number of exceedances for high thresholds will often be zero and since ranks are only discarded from the histogram if all ensemble members and the verification field have the same FTE, the FTE histogram for very high thresholds will be composed largely

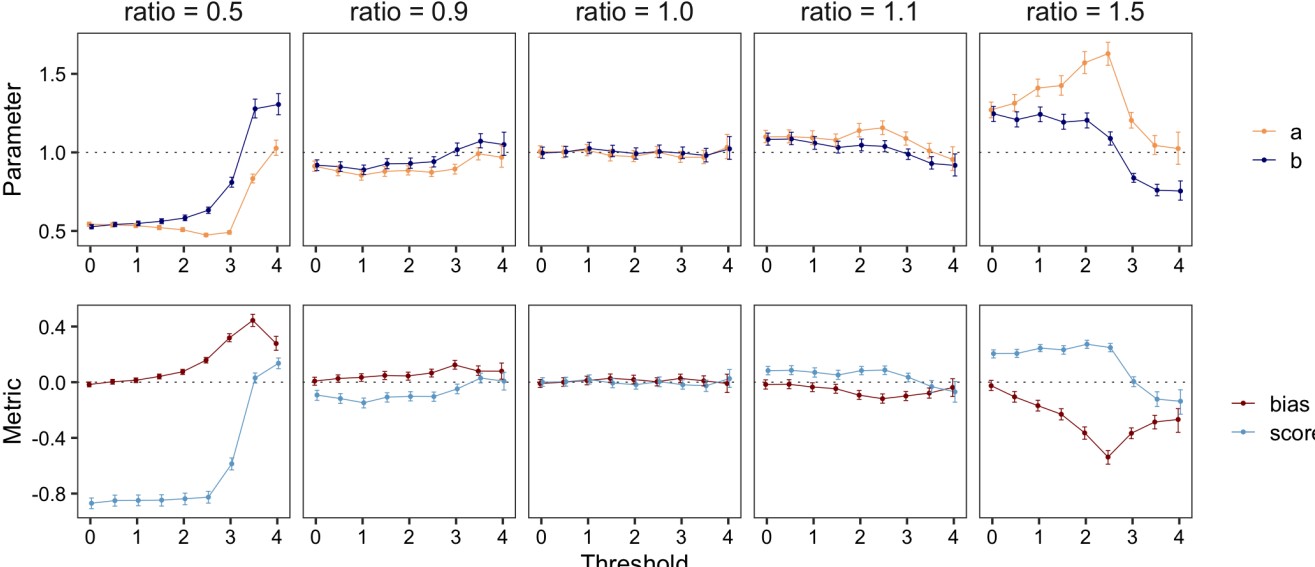

**Figure 5.** Estimated beta distribution parameters (top) and corresponding $\beta$-score and $\beta$-bias (bottom) of FTE histograms calculated over different thresholds for forecasts with low, even, and high correlation length ratios about $a_0 = 2$. Vertical lines denote the 95% confidence interval found via the nonparametric bootstrap method.

of ranks resulting from ties broken uniformly at random. This results in a more deceptively uniform histogram which explains the tendency toward zero. For the most extreme thresholds studied here, the confounding effect of resolving ties (which can exist between all but a single ensemble member FTE) at random becomes very dominant, and histogram shapes get distorted to a degree where the interpretations provided in Table 1 no longer hold. The associated FTE histograms still exhibit non-uniformity and thus indicate that the ensemble forecasts are not perfectly calibrated, but it becomes impossible to diagnose the particular type of miscalibration from the histogram shape. We can also see that the sampling variability increases with increasing threshold since more and more uninformative cases with fully tied FTE values exist, so a much larger total number of verification cases is required in order to have a comparable number of informative cases. In practice, if FTE histograms are used as a diagnostic tool, we recommend focusing on moderate thresholds. If they are used to compare the calibration of different forecast systems, they can still be effective at more extreme thresholds.

Another variable of interest in evaluating the FTE histograms is the size of the domain to which the metric is applied. In our simulation framework, making the domain larger or smaller while keeping the correlation length constant is equivalent to keeping the domain size constant and varying the correlation length of the verification field. That is, for a fixed domain size, a smaller correlation length mimics a "large domain" (with low resolution) and larger correlation length mimics a "small domain" (with high resolution). Analyzing the $\beta$-score and $\beta$-bias over a range of correlation length ratios is then equivalent to studying the FTE histograms' utility for different domain sizes. For the domain used in this study, a correlation length of 1

is considered small and 3 is considered large (see Fig. 2). In either case, Fig. 6 shows that the $\beta$-score quickly deviates from
zero when the correlation length *ratio* is different from one. The $\beta$-score's relatively steep slope around the correlation length
of 1.0 in both cases indicates that the FTE histogram maintains good discrimination ability regardless of domain size, provided
that there are sufficiently many grid points within the domain to keep the (spatial) sampling variability associated with the
calculation of the FTE values low. Notably, the inverse relationship between $\beta$-bias and the correlation length ratio is also in
agreement with Table 1.

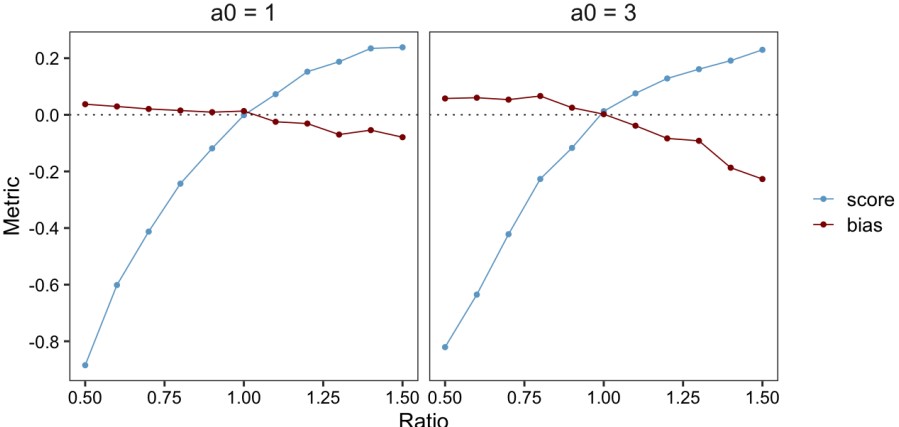

**Figure 6.** Estimated $\beta$-score and $\beta$-bias of FTE histograms constructed with $\tau = 1$ for verification correlation length $a_0 \in \{1, 3\}$ and ensemble range $a_M$ varying from $0.5a_0$ to $1.5a_0$.

We now turn attention back to our motivating figure (Fig. 1), which was created with verification correlation length $a_0 = 2$.
Forecast 1 exhibited the correct spatial structure (i.e., $a_M = 2$), forecast 2 was incorrectly specified with correlation length $a_M$
= 3, and forecast 3 was incorrectly specified with correlation length $a_M$ = 1.8. While it may be obvious that forecast 2 has
incorrect spatial structure, the structural difference between forecasts 1 and 3 is not so apparent. However, as demonstrated by
the analysis above (Fig. 2 – Fig. 5), these misspecifications are certainly identifiable using FTE histograms.

## 4   Application to downscaling of ensemble precipitation forecasts

Distributed hydrological models like NOAA's National Water Model (NWM) require meteorological inputs at a relatively high
spatial resolution. At shorter forecast lead times (typically up to one or two days ahead) limited-area NWP models provide
such high-resolution forecasts, but for longer lead times only forecasts from global ensemble forecast systems like NOAA's
GEFS are available. These come at a relatively coarse resolution and need to be downscaled (statistically or dynamically) to
the high-resolution output grid. Here, we use a combination of the statistical post-processing algorithm proposed by Scheuerer
and Hamill (2015), ensemble copula coupling (ECC; Schefzik et al., 2013), and the spatial downscaling method proposed by
Gagnon et al. (2012) to obtain calibrated, high-resolution precipitation forecast fields based on GEFS ensemble forecasts. Does

the spatial disaggregation method produce precipitation fields with appropriate sub-grid scale variability? This question will be answered using the FTE-based verification metric discussed above.

## 4.1 Data and downscaling methodology

We consider 6-hour precipitation accumulations over a region in the South-Eastern US between -91° and -81° longitude and 30° and 40° latitude during the period from January 2002 to December 2016. Ensemble precipitation forecasts for lead time 66-h to 72-h were obtained from NOAA's second-generation GEFS reforecast dataset (Hamill et al., 2013) at a horizontal resolution of ∼0.5°. Downscaling and verification is performed against precipitation analyses from the ∼0.125° climatology-calibrated precipitation analysis (CCPA) dataset (Hou et al., 2014).

In order to obtain calibrated ensemble precipitation forecasts at the CCPA grid resolution we proceed in three steps. First, we apply the post-processing algorithm by Scheuerer and Hamill (2015) to the GEFS forecasts and upscaled (to the GEFS grid resolution) precipitation analyses in order to remove systematic biases and ensure adequate representation of forecast uncertainty at this coarse grid scale. The resulting predictive distributions are turned back into an 11-member ensemble using the ECC-mQ-SNP variation (Scheuerer and Hamill, 2018) of the ECC technique. This variation removes discontinuities and avoids randomization that can occur when the standard ECC approach is applied to precipitation fields. Finally, each ensemble member is downscaled from the GEFS to the CCPA grid resolution using a slightly simplified version of the Gibbs sampling disagregation model (GSDM) proposed by Gagnon et al. (2012). To generate downscaled fields with spatial properties that vary depending on the season, we rely here on a monthly calibration of the GSDM, rather than on meteorological predictors as in the original model. The 15 years of data are cross-validated: one year at a time is left out for verification and the post-processing and downscaling models are fitted with data from the remaining 14 years. Repeating this process for all years leaves us with 15 years of downscaled ensemble forecasts and verifying analyses. See Fig. 7 for a visual reference.

## 4.2 Univariate verification

Before applying the FTE histogram to investigate whether the spatial disaggregation used in the downscaling method produces precipitation fields with appropriate sub-grid scale variability, we check the calibration of the univariate ensemble forecasts across all fine scale grid points. We study (separately) the months January, April, July, and October in order to represent winter, spring, summer, and fall, respectively. Daily analyses and corresponding ensemble forecasts from each of these months are pooled over the entire verification period and all grid points within the study area, and are used to construct the verification rank histograms in Fig. 8. Cases where all ensemble member forecasts and the analysis are tied – for example, when there is zero accumulation at a grid point for all fields – are withheld from the histogram to avoid artificial uniformity introduced by breaking ties in rank at random.

Ideally, the statistical post-processing and downscaling should yield calibrated ensemble forecasts and thus rank histograms that are approximately uniform. Clearly, the rank histograms for the downscaled forecast fields shown in Fig. 8 are not uniform; there is a consistent peak in the higher ranks indicating that the downscaled ensemble forecasts tend to underestimate precipitation accumulations, especially in fall and winter. This bias could be an indication that either the post-processing distribution

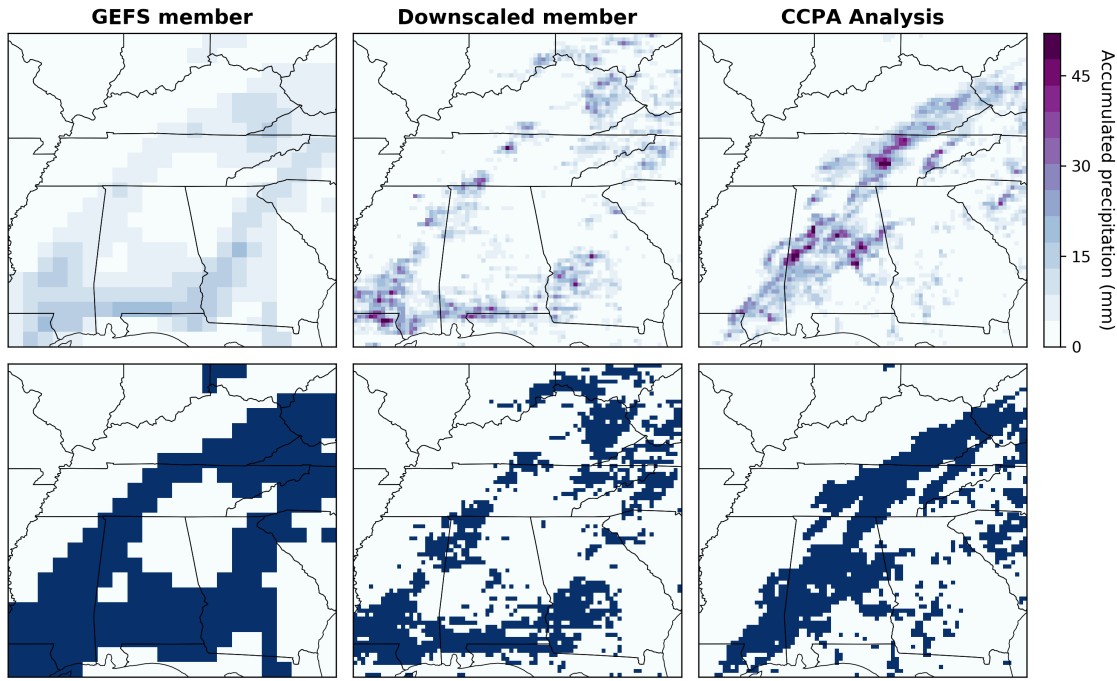

**Figure 7.** Examples of different data fields for 6-hour precipitation accumulation on July 24, 2004 (top) and corresponding 5mm binary exceedance fields (bottom; dark blue regions indicate threshold exceedance). From left to right: coarse-scale GEFS ensemble member, the same member downscaled to the analysis resolution, and the corresponding CCPA analysis.

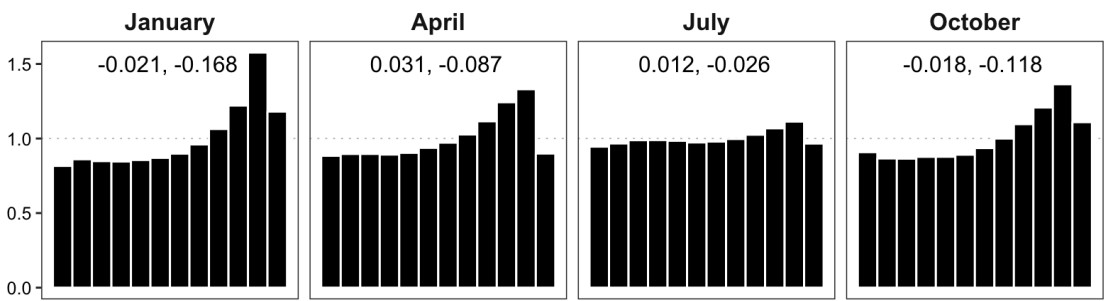

**Figure 8.** Verification rank histograms (density) for downscaled fields at representative months with cases of fully tied ranks removed. Estimated $\beta$-score (left) and $\beta$-bias (right) annotated.

(gamma) or the disaggregation distribution (log-normal) are not perfectly suited to represent the respective forecast uncertainties. It may also be a result of a superposition of biases in different sub-domains or for different weather situations. Univariate calibration in July – which happens to be a month with more frequent precipitation in this region of the US – is relatively good, and while the histograms of other months show clear departures from uniformity, there is at least no strong ∪– or ∩–shape to

indicate significant dispersion errors. We thus continue with our analysis of the spatial calibration of the downscaled ensemble forecast fields, keeping in mind though that the under-forcast biases seen in Fig. 8 will carry over to the FTE histograms, and will superimpose any shape resulting from spatial miscalibration. In July, where only a mild under-forcast bias is observed, we will have the best chance of drawing unambiguous conclusions about the spatial structure from the shape of the FTE histogram.

### 4.3 Verification of spatial structure

In the remaining analysis, we employ FTE histograms to investigate the spatial properties of the ensemble forecast fields obtained by the downscaling algorithm for the same representative months outlined above. Spatial variability of precipitation fields depends on whether precipitation is stratiform or convective, and in the latter case also on the type of convection (local vs synoptically forced). The frequency of occurrence of these categories has a seasonal cycle, and it is therefore interesting to study how well the downscaling methodology works in different seasons. The first step in computing the FTE is deciding what value to use for the threshold. If the climatology varies strongly across the domain, it may be desirable to use a variable threshold such as a climatology percentile. However, the South-Eastern US is a flat and relatively homogeneous region meaning the precipitation accumulation patterns will not be affected as much by orography, and we therefore select a fixed threshold for constructing FTE histograms. Another advantage of this approach is that a fixed threshold has a direct physical interpretation; here we use thresholds of 5mm, 10mm, and 20mm to study the spatial calibration of the ensemble for low, medium, and high accumulation levels over the 6-hour window.

In Fig. 9, it is clear by visual inspection that the FTE histograms are all ∪-shaped to some extent, though the corresponding $\beta$-scores highlight that the histograms are explicitly more uniform in the fall and winter months. In the spring and summer months (i.e., April and and July) the histograms reveal a clear under-dispersion in the ensemble FTEs at all analyzed thresholds. This would suggest that the downscaled ensemble overestimates fine scale variability during the seasons with more convective events. This could indicate that the calibration procedure of the GSDM downscaling method in Gagnon et al. (2012) struggles with selecting good parameters that produce downscaled precipitation fields with just the right amount of spatial variability during the summer season with mainly (but not exclusively) convective precipitation. The FTE histogram can thus provide valuable diagnostic information that helps identify shortcomings of a forecast methodology. Indeed, in one of our current projects we seek to improve the GSDM, with one objective being to calibrate the model such that the downscaled fields reproduce the correct amount of spatial variability, in a flow-dependent fashion, using meteorological predictors such as instability indices and vertical wind shear (Bellier et al., 2020). For the $\beta$-biases seen in Fig. 9, the interpretation is more difficult. Their value at higher thresholds has the opposite sign as what we would expect from Table 1 in a situation where the forecast fields simulate excessive fine scale variability. As noted above, however, this is likely due to an under-forecast bias in the marginal distributions and the associated effect on the $\beta$-bias which is opposite to (and seems to be dominating) the effects of spatial miscalibration.

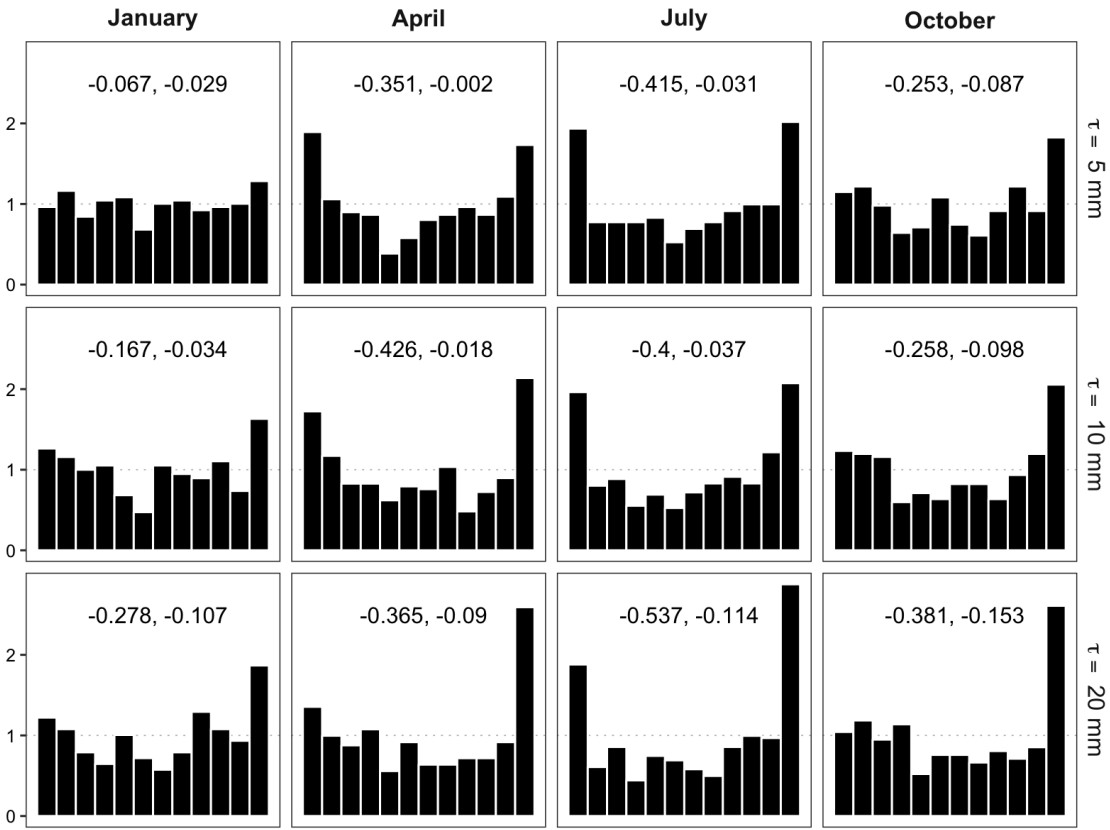

**Figure 9.** FTE histograms for downscaled fields at different thresholds in representative months. Estimated $\beta$-score (left) and $\beta$-bias (right) annotated.

## 5   Conclusions

When forecasting meteorological variables on a spatial domain, it is important for many applications that not only the marginal forecast distributions but also the spatial (and/or temporal) correlation structure is represented adequately. In some instances, misrepresentation of spatial structure by ensemble forecast fields may be visually obvious; otherwise, a quantitative verification metric is desired to objectively evaluate the ensemble calibration. The FTE metric studied here is a projection of a multivariate quantity (i.e., a spatial field) to a univariate quantity, and can be combined with the concept of a (univariate) verification rank histogram to analyze the spatial structure of ensemble forecast fields. This idea has first been applied by Scheuerer and Hamill (2018) to study the properties of downscaled ensemble precipitation forecasts, but an understanding of the general capability of the FTE metric to detect misrepresentation of the spatial structure by the ensemble has been lacking as yet.

In this paper, we performed a systematic study in which we simulated ensemble forecast and verification fields with different correlation lengths to understand how well a misspecification of the correlation length can be detected by the FTE metric. To

this end, the metric was slightly extended and is composed of three steps: (1) calculate the FTE of each verification and ensemble forecast field, (2) construct an FTE histogram over available instances of forecast and verification times, and (3) derive the $\beta$-score and $\beta$-bias from the stochastically disaggregated FTE histogram to characterize departure from uniformity. We have found that the FTE metric is capable of detecting even minor issues with the correlation length (e.g., 10% miscalibration) in ensemble forecasts, and this conclusion was consistent across a range of thresholds and domain sizes. Applied in a data example with downscaled precipitation forecast fields, the FTE metric pointed to some shortcomings of the underlying spatial disaggregation algorithm during the seasons where precipitation is driven by local convection.

The FTE metric is relatively simple and enjoys an easy and intuitive interpretation. In particular, the $\beta$-score and $\beta$-bias can be compared according to Table 1 to diagnose shortcomings in the calibration of ensemble forecasts. If different types of miscalibration occur together, additional diagnostic tools like univariate verification rank histograms have to be considered alongside the FTE histograms to disentangle the different effects. While we have focused on histograms in the analysis of the verification FTE rank here, the same projection could also be used in combination with proper scoring rules. We believe that FTE histograms are a useful addition to the set of spatial verification metrics. They complement metrics like the wavelet-based verification approach proposed by Buschow et al. (2019) which has additional capabilities when it comes to analyzing aspects of the spatial texture of forecast fields but is not primarily targeted at proper uncertainty quantification by an ensemble.

*Code and data availability.* The code and data used for this study are available in the accompanying Zenodo repositories (code: https://doi.org/10.5281/zenodo.3945515, data: https://doi.org/10.5281/zenodo.3945512). The most recent version of the code can also be found in JJ's Git repository (https://github.com/joshhjacobson/FTE).

## Appendix A: Stochastic disaggregation of transformed ranks

The data vector of ranks $\mathbf{r}$ has discrete elements $r_i \in \{1, 2, \ldots, k+1\}$ where $k$ is the number of ensemble members. In order to disaggregate these elements to a continuous domain for use with maximum likelihood estimation, the following algorithm is applied to each element $r_i$:

1. Let $d_i = (r_i - \frac{1}{2})/(k+1)$.

2. Simulate a (continuous) uniform random variable

$$U_i \sim Uniform\left(d_i - \frac{1}{2(k+1)}, d_i + \frac{1}{2(k+1)}\right)$$

3. Set $r_i = U_i$.

The effect of Step 1 is a mapping into $[0, 1]$, while Step 2 is the stochastic disaggregation to evenly-spaced uniform intervals whose supports form a partition of unity of $[0,1]$.

## Appendix B: Properties of simulated ensemble members

Let $Z_M(s)$ and $W_i(s)$ be independent, mean-zero Gaussian processes, each with Matérn covariance function $M(d|\nu_M, a_M)$. Now suppose $Z_M$ and $W_i$ are independent standard Gaussian random variables representing the marginal distribution of processes $Z_M(s)$ and $W_i(s)$. Setting random variable

$$Z_i = \omega Z_M + \sqrt{1 - \omega^2} W_i, \quad \omega \in [-1, 1], \tag{B1}$$

we see $\mathbb{E}[Z_i] = 0$ by linearity of the expectation operator, and

$$\begin{aligned} \text{Var}[Z_i] &= \text{Var}[\omega Z_M + \sqrt{1 - \omega^2} W_i] \\ &= \omega^2 \text{Var}[Z_M] + (1 - \omega^2) \text{Var}[W_i] \\ &= 1 \end{aligned} \tag{B2}$$

using independence of $Z_M$ and $W_i$. Then $Z_i$ is a standard Gaussian random variable representing the marginal distribution of ensemble member $Z_i(s) = \omega Z_M(s) + \sqrt{1 - \omega^2} W_i(s)$. Further, observe that

$$\begin{aligned} \text{Cov}&[Z_i(s+h), Z_i(s)] \\ &= \text{Cov}[\omega Z_M(s+h) + \sqrt{1 - \omega^2} W_i(s+h), \omega Z_M(s) + \sqrt{1 - \omega^2} W_i(s)] \\ &= \omega^2 \text{Cov}[Z_M(s+h), Z_M(s)] + (1 - \omega^2) \text{Cov}[W_i(s+h), W_i(s)] \\ &= \omega^2 M(||h|| | \nu_M, a_M) + (1 - \omega^2) M(||h|| | \nu_M, a_M) \\ &= M(||h|| | \nu_M, a_M) \end{aligned} \tag{B3}$$

by independence of $Z_M(s)$ and $W_i(s)$. That is, ensemble members $Z_i(s)$, $i = 1, \ldots, k$, preserve the covariance structure of the ensemble mean $Z_M(s)$.

## Appendix C: Derivation of an appropriate co-located correlation coefficient

Suppose $Z_0$ and $Z_M$ are standard Gaussian random variables with $\text{Corr}[Z_0, Z_M] = \rho$, and $\{W_i\}_{i=1}^k$ is a set of independent standard Gaussian random variables, each independent of $Z_0$ and $Z_M$. Define

$$Z_i = \omega Z_M + \sqrt{1 - \omega^2} W_i, \quad i = 1, \ldots, k, \quad \omega \in [-1, 1]. \tag{C1}$$

From Appendix B we have that each of $Z_i$ is again a standard Gaussian random variable. Then, for $i \neq j$, we see that

$$\begin{aligned} \text{Cov}[Z_i, Z_j] &= \text{Cov}(\omega Z_M + \sqrt{1 - \omega^2} W_i, \omega Z_M + \sqrt{1 - \omega^2} W_j) \\ &= \omega^2 \text{Cov}(Z_M, Z_M) \\ &= \omega^2 \end{aligned} \tag{C2}$$

using pairwise independence of $Z_M, W_i$ and $W_j$. Using a similar technique we see

$$\text{Cov}[Z_0, Z_i] = \omega\rho. \tag{C3}$$

Now let $\mathbf{Z} = (Z_0, Z_1, \ldots, Z_k)'$. Then,

$$\text{Cov}[\mathbf{Z}] = \begin{pmatrix} 1 & \omega\rho & \ldots & \ldots & \omega\rho \\ \omega\rho & \ddots & \omega^2 & \ldots & \omega^2 \\ \vdots & \omega^2 & \ddots & \ddots & \vdots \\ \vdots & \vdots & \ddots & \ddots & \omega^2 \\ \omega\rho & \omega^2 & \ldots & \omega^2 & 1 \end{pmatrix} \tag{C4}$$

Setting $\rho = \omega$ is thus necessary (except for the trivial case where $\omega = 0$) and sufficient for univariate probabilistic calibration of the ensemble as this choice makes $Z_0$ indistinguishable from $Z_1, \ldots, Z_k$ in distribution.

## Appendix D: Sensitivity of FTE histograms to forecast skill

The simulation setup introduced in section 3.2 allows us to control the skill of the synthetic ensemble forecasts through the parameters $\rho$ and $\omega$, where (as shown above) the restriction $\rho = \omega$ is required to ensure calibration of the marginal distributions.

Does the sensitivity of the FTE histogram to mis-specified correlation lengths change with changing forecast skill? To investigate this further, we show results for the extreme 'no skill' case where $\rho = \omega = 0$, to complement those shown above where we simulated forecasts with a relatively high correlation ($\rho = \omega = 0.8$) between ensemble mean and verification at each grid point. The other parameters remain unchanged; i.e., simulation experiments are performed for $a_0 = 2$ and $a_M \in \{1, 1.8, 2, 2.2, 3\}$. By Eq. (8), the ensemble members are then independent realizations of a mean zero Gaussian random field with Matérn covariance

having correlation length $a = a_M$.

Figure D1 gives an example of simulated ensemble member fields which are mutually independent and uncorrelated with the verification field, while their spatial structure is 10% miscalibrated in the case of rows A and C. In contrast to Fig. 3, where the positive skill of the ensemble forecasts entails some degree of correspondence between the features in the forecast and verification fields, there is no such correspondence in between the fields in Figure D1. The corresponding FTE histograms and

their associated $\beta$-scores, however, are able to identify row B as spatially calibrated and row A (row C) as having a correlation length ratio that is too small (large).

A similar story is provided by Fig. D2 which, like Fig. 5 where $\rho = \omega = 0.8$, demonstrates how the estimated $\beta$-distribution parameters, $\beta$-score, and $\beta$-bias vary over increasing thresholds for different correlation length ratios. While the associated experiments differ in that ensemble members have no univariate skill here (i.e., $\rho = \omega = 0$), the behavior witnessed in the two

figures is nearly indistinguishable. Thus, we conclude that correlation between the ensemble and verification has a negligible effect on the FTE histograms' ability to detect miscalibration in spatial structure. This was not obvious to us a priori, but perhaps one can think of these 'no skill' simulations as the residual fields that remain after the 'predictable component' has

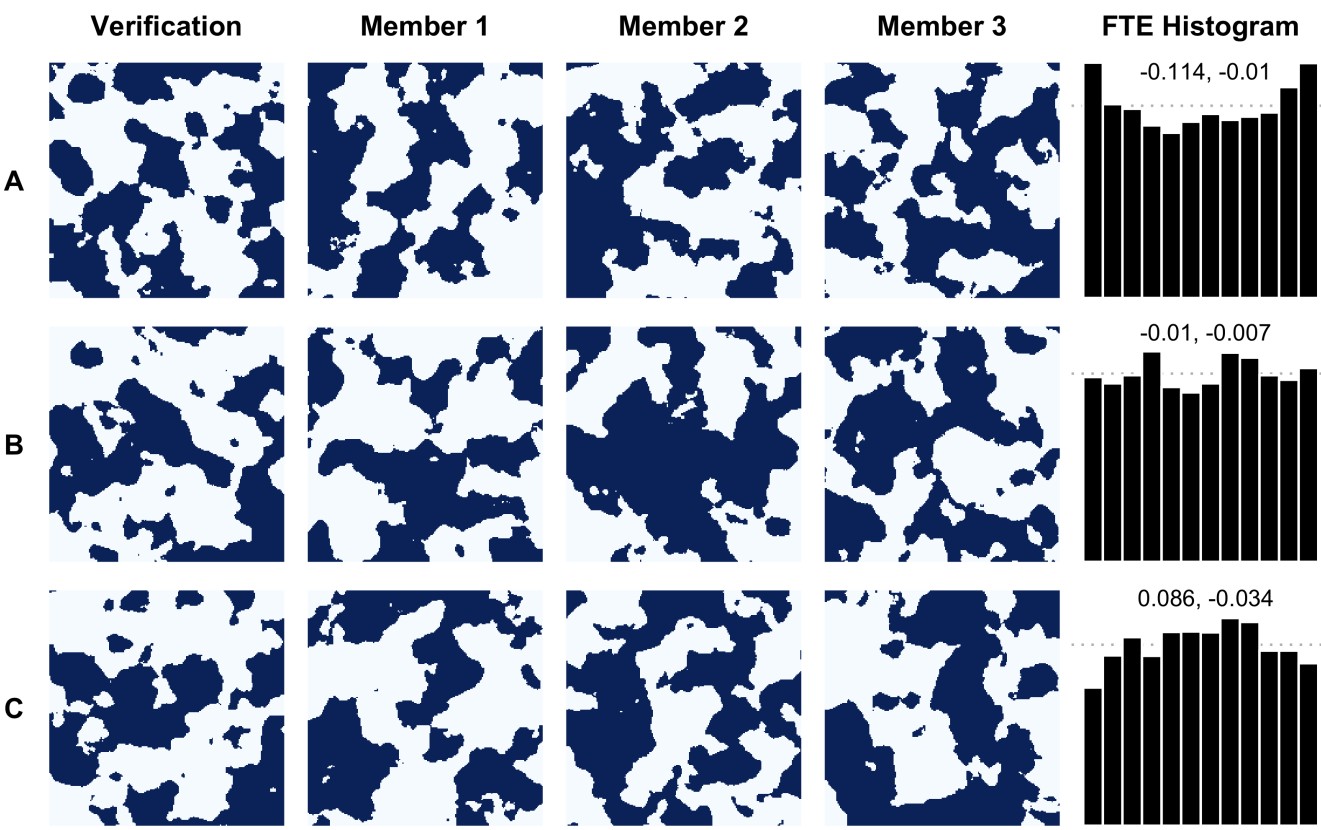

**Figure D1.** As Fig. 3, but ensemble fields are constructed with skill parameter $\rho = \omega = 0$.

been subtracted from both ensemble member and verification fields. In any case, the insensitivity to forecast skill is good
news for the practical application of FTE histograms, where forecast skill is usually unknown, and confounding effects are
undesirable. It is a reminder though that they are a tool for assessing forecast calibration, not forecast skill.

*Author contributions.* This study is based on the Master's work of JJ, under supervision of WK and MS. The concept of this study was
developed by MS and extended upon by all involved. JJ implemented the study and performed the analysis with guidance from WK and MS.
JB provided the downscaled GEFS forecast fields. JJ, WK, and MS collaborated in discussing the results and composing the manuscript,
with input from JB on section 4.

*Competing interests.* The authors declare that they have no conflict of interest.

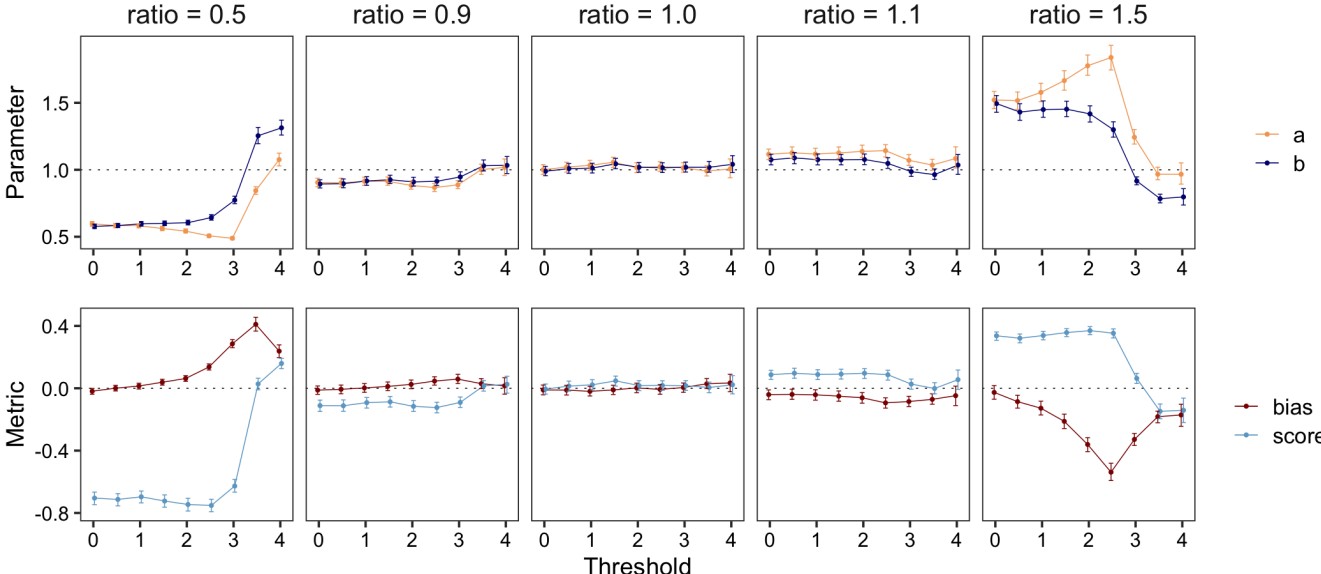

**Figure D2.** As Fig. 5, but ensemble fields are constructed with skill parameter $\rho = \omega = 0$.

*Acknowledgements.* JJ was supported by NSF DMS-1407340. WK was supported by NSF DMS-1811294 and DMS-1923062. MS was supported by funding from NOAA/ESRL/PSD. JB was supported by a funding agreement between NOAA/ESRL/PSD and NOAA/NWS/MDL on the development and transfer of advanced statistically postprocessed probabilistic ensemble guidance. This work utilized resources from the University of Colorado Boulder Research Computing Group, which is supported by the NSF (awards ACI-1532235 and ACI-1532236), the University of Colorado Boulder, and Colorado State University.

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
