# Peer review of "Beyond Univariate Calibration: Verifying Spatial Structure in Ensembles of Forecast Fields"

_Nonlinear Processes in Geophysics, 2019_

## Referee Comment (RC1) · Anonymous Referee #1 · 3 Feb 2020

General comments:

The authors study a recently developed extension of the well-known analysis rank histogram, which allows for a verification of spatial structures in ensemble forecasts. A simulation study using Gaussian random fields serves to demonstrate the connection between spatial correlation lengths and the rank histogram of the fraction of threshold exceedance (FTE). As a real world example application, the authors study statistically downscaled ensemble predictions of precipitation accumulations. The presented verification technique is interesting and could be useful to a potentially wide range of users, thanks to its relative simplicity and intuitive connection to the analysis rank histogram.

The manuscript is well written and overall easy to follow, the experiments adequately demonstrate the merits of the FTE histograms. My comments and questions mostly concern the analysis of the results. When these points have been addressed, I recommend publication of the manuscript.

1. I'm not sure that the interpretation of the FTE is always histogram quite as straightforward as you say: For the Gaussian random fields studied in section 3.3, larger correlation lengths always lead to a greater range of FTEs in the ensemble, but is this necessarily true in the general case? While I cannot, off the top of my head, name a random process that violates this assumption, I could imagine a weather prediction system which has been tuned to produce correct rain areas but systematically simulates patterns with too many small showers or too broad stratiform rain fields. In such cases, the spatial correlation structure would be misrepresented in a way that cannot be detected from the area above a given threshold alone.

2. If you follow Keller and Hense (2011) and fit a beta distribution to the histograms, why not also use their beta-score which combines a and b into a single number that characterizes over- and under-dispersion? I fail to see the advantage of considering a and b separately.

3. Related to the previous point, please explain more clearly (preferably in section 2) how you interpret the skewed cases a>b and b>a. While the skewness certainly relates to an overall over- or under-estimation of exceedance areas, maybe a more careful discussion is needed: Keller and Hense define a "beta-bias" as b-a, which is similar but not identical to the ensemble mean bias.

4. Can you comment more on the case where the margins are mis-calibrated? This is not covered by your simulation study, but the effects could potentially be a major drawback of you method. How can you be certain that the FTE really indicates errors in the spatial structure and is not determined by the erroneous marginal distribution? Could you remove the effect of the margins before calculating the FTE-histogram?

Specific comments:

page 1, l.6f: why not say exactly how the FTE is defined in the abstract? The idea is not so complicated and can be explained without equations.

page 2, l.44f: you might also cite Kapp et al. (2018) "Spatial verification of high-resolution ensemble precipitation forecasts using local wavelet spectra", who also use wavelets but specifically only look at ensemble predictions. They also project the fields to a lower-dimensional space and then apply standard multivariate verification techniques (similar to your approach), but the interpretation of their results is not so straightforward in terms of correlation lengths.

page 4, l.93f: please explain the relationship between the correlation length and the shape of the histogram in a little more detail. I think this point is pretty central and not completely obvious.

page 4, l.97f: are you sure that cases where forecast and observations never exceed the threshold should just be discarded entirely? Shouldn't the addition of a few correct negative cases to the verification data-set lead to a better score for the prediction system?

page 5, table 1: Consider adding a third column which states the interpretation in terms of the correlation length or the quality of the forecast. However, I'm not sure what exactly to write for ab (see main comment 3).

page 5, l.125: If I understand correctly, the first step in this equation is only correct if F is a strictly monotonic function. What if the distribution contains a point mass, say at zero as for precipitation? Does that affect the rest of you argumentation? More generally, can you comment on how the results of your simulation study transfer to discontinuous variables like precipitation (as studied in section 4)?

page 5, l.132: Do forecast and verification really need to be correlated for this experiment? What would happen if they weren't?

page 6, l.139: You should make it clear whether s and h are vectors or scalars, a process where C depends on a vector valued h would be anisotropic.

page 6, l.147f: I don't understand the grammar in the sentence starting with "The multivariate ...", is "Matern" the subject and "sets" the predicate?

page 7, l.184: Please clarify in what sense these fields are "realistic".

page 8, Fig.2: Please make it unambiguous in the figure or caption which parameter value is a and which is b.

page 9, l.205: You should discuss the origin and interpretation of this skewness (see general comment 3). It looks like forecasts whose only shortcoming lies in the correlation length obtain a left or right skew at certain thresholds (Fig. 5). What does that mean for the interpretation of a>b and a<b in realistic situations?

page 10, Fig.5: What's a0 here?

page 11, l220: At least for the two most extreme ratio, a and be not only tend to one but go beyond and indicate the opposite bias. How do you interpret that? Can you maybe add error bars to these plots (via bootstrapping?) to check whether some of these effects might be insignificant?

page 11, l221: While it is true that longer correlation lengths on the same grid lead to the same patterns as a decrease in domain size, are you sure that the number of available grid-points has no effect at all? Surely for very small domains it becomes increasingly hard to confidently estimate the spatial correlation structure?

page 13, l272f: why not quantify the shape of the univariate histograms via a beta-distribution as well?

page 14, l294f: Can you discuss the skewness seen in Fig.9? The difference between a and b increases with threshold but, if I understand correctly, has a different sign than seen in Fig.5 ( a>b but both are smaller than 1).

page 14, l297f: I would also note that this kind of consideration could generally help you distinguish the effects of miscalibrated margins from errors in the spatial structure: If the miscalibration seen in figure 9 were caused by errors in the marginal distributions, one might expect the effect to be weakest in July were the marginal calibration was the best.

page 15, Fig.9: Some of these histograms are clearly not uni-modal. Can you comment on how your method is affected by cases where the beta-fit is likely not very good?

---

## Referee Comment (RC2) · Anonymous Referee #2 · 17 Jun 2020

This manuscript discusses multivariate verification of ensemble forecasts with a focus on spatial structures. A new approach is proposed, illustrated and discussed using synthetic and precipitation datasets.

The proposed approach consists in transforming a multivariate quantity (a spatial field) into a univariate quantity. This is performed by thresholding and counting the fraction of grid points exceeding a threshold over a domain. The popular rank histogram tool is applied to the resulting ensemble and observed fractions of threshold exceedance (FTE). The method is easy to implement and is appealing because of the simplicity of both the interpretation of the derived univariate product and of the derived rank

histograms.

In order to better understand FTE histograms, a toy model is used to assess the sensitivity of the method to predefined discrepancies in a controlled environment. In a real-life example context, FTE histograms are derived for precipitation forecasts and limitations of the underlying forecasting system diagnosed. The paper is well-written, well-structured, and pleasant to read. The description of the method and datasets is clear, the discussion of the results convincing.

However, the manuscript would benefit from some clarifications. First, the authors could clarify the link between the FTE histogram approach and the fraction skill score (FSS, Roberts and Lean 2008), a popular verification metric applied to deterministic forecast of precipitation fields. The first step is similar for the two methods (transformation of a real-value field into a binary field and then a focus on the fraction of events over a domain). It would be beneficial for the verification community to clearly state the link between FTE histogram and FSS.

Secondly (and more importantly), the author could discuss the impact of miscalibration of the univariate distribution on the interpretation of the FTE. It is cleared mentioned in the text that this aspect should not be disregarded. In line 86, it is stated that the FTE histogram can be used "once the marginal distribution has been checked", and in Section 4.2 that "the possible non-uniformity of FTE histograms [...] could be due to univariate miscalibration". Perfect calibration of univariate distributions cannot be expected in reality so a discussion about the permeability of FTE histograms to univariate miscalibration would be more than useful for the interpretation of real case FTE histograms.

In addition, the authors could clarify why the parameters derived from the beta distribution fitting differ from the parameters defined in the Keller and Hense 2011 paper. The list of references could also be checked and the duplicated doi information removed.

Reference: Roberts NM, Lean HW. 2008. Scale-selective verification of rainfall accumulations from high-resolution forecasts of convective events. Mon. Weather Rev. 136: 78– 97.

---

## Author Comment (AC1) · 15 Jul 2020

We appreciate the time and energy these two reviewers have put into their thoughtful comments – they have surely helped to strengthen the findings in this manuscript. Below, we address each comment with our response given in italics. We have also provided a PDF highlighting the implemented changes as a supplement to this reply.

Kind regards,

Joshuah Jacobson, William Kleiber, Michael Scheuerer, and Joseph Bellier

[Figure]

**Reviewer 1**

General comments:

The authors study a recently developed extension of the well-known analysis rank histogram, which allows for a verification of spatial structures in ensemble forecasts. A simulation study using Gaussian random fields serves to demonstrate the connection between spatial correlation lengths and the rank histogram of the fraction of threshold exceedance (FTE). As a real world example application, the authors study statistically downscaled ensemble predictions of precipitation accumulations. The presented verification technique is interesting and could be useful to a potentially wide range of users, thanks to its relative simplicity and intuitive connection to the analysis rank histogram. The manuscript is well written and overall easy to follow, the experiments adequately demonstrate the merits of the FTE histograms. My comments and questions mostly concern the analysis of the results. When these points have been addressed, I recommend publication of the manuscript.

1. I'm not sure that the interpretation of the FTE histogram is always quite as straightforward as you say: For the Gaussian random fields studied in section 3.3, larger correlation lengths always lead to a greater range of FTEs in the ensemble, but is this necessarily true in the general case? While I cannot, off the top of my head, name a random process that violates this assumption, I could imagine a weather prediction system which has been tuned to produce correct rain areas but systematically simulates patterns with too many small showers or too broad stratiform rain fields. In such cases, the spatial correlation structure would be misrepresented in a way that cannot be detected from the area above a given threshold alone.

*Our simulated fields try to simulate this effect with small correlation length corresponding to small showers and large correlation lengths corresponding to stratiform precipitation. For example, notice how the spatial patterns in the simulated binary exceedance fields of Figure 2 loosely resemble that of the newly added 5mm binary exceedance data fields in Figure 7. We agree that Gaussian random fields do not provide a comprehensive study of all possible spatial patterns that we might expect to see in practice, but for the purposes of exploring / understanding the FTE histogram behavior in an idealized setting we believe the study is appropriate.*

2. If you follow Keller and Hense (2011) and fit a beta distribution to the histograms, why not also use their beta-score which combines a and b into a single number that characterizes over- and under-dispersion? I fail to see the advantage of considering a and b separately.

   *Thank you for this suggestion. We agree that the beta-score and beta-bias, as described in Keller and Hense (2011), are more straightforward and interpretable for characterizing over- and under-dispersion than the estimated beta-parameters alone. As such, we have updated our discussion, Table 1, and all relevant figures to focus interpretation of the FTE histogram through the lense of these metrics.*

3. Related to the previous point, please explain more clearly (preferably in section 2) how you interpret the skewed cases a>b and b>a. While the skewness certainly relates to an overall over- or under-estimation of exceedance areas, maybe a more careful discussion is needed: Keller and Hense define a "beta-bias" as b-a, which is similar but not identical to the ensemble mean bias.

   *If the marginal distributions are mis-calibrated, over/under-estimation of the observed values directly translates into skewed FTE-histograms. If the marginal distributions are well-calibrated but spatial correlations are over/underestimated, skewness and cap/cup-shape of the FTE histogram are somewhat intertwined,*

*depending on the threshold value. To see this, consider the extreme case where the ensemble forecasts are spatially uncorrelated while the observations are fully correlated. If the threshold is set to the climatological median at each grid point, the FTE for each ensemble member is close to 0.5 while the FTE for the observed field is either 0 or 1, each with equal probability. So the FTE histogram is cup-shaped with half of the cases in the lowest bin and the other half in the highest bin. If we pick a larger threshold, say the 0.95 climatological quantile, the FTE of all ensemble forecasts is close to 0.05, while the FTE of the observed field is 1 with probability 0.05 and 0 with probability 0.95. The associated FTE histogram will thus still have all cases concentrated in the lowest and the highest bin, but more (95%) cases in the lower bin, so the cup shape goes along with a skew. In reality, spatial correlations won't be at the two extreme ends (full or no spatial dependence), but we can expect (and we see in the simulations) a similar, threshold-dependent link between cap/cup shape and skewness of FTE histograms. We have expanded our explanations regarding the link between spatial miscalibration and histogram shapes (beta parameters) such that it now includes the skewed cases.*

4. Can you comment more on the case where the margins are mis-calibrated? This is not covered by your simulation study, but the effects could potentially be a major drawback of your method. How can you be certain that the FTE really indicates errors in the spatial structure and is not determined by the erroneous marginal distribution? Could you remove the effect of the margins before calculating the FTE-histogram?

*The case of mis-calibrated margins is indeed a challenge, and we added a discussion of this situation to section 2. Over- and under-forecast biases can be removed quite easily, but for dispersion errors this is not straightforward. And it is true that the effects of marginal mis-calibration and spatial mis-calibration are hard to disentangle, and that this hampers the interpretation of FTE histograms.*

*We note though (and have added a similar statement to the manuscript) that this drawback is not specific to our approach but applies to any multivariate verification metric that is based on the idea of projecting a multivariate quantity onto a univariate one (as almost all of them do in order to condense information).*

Specific comments:

- page 1, l.6f: why not say exactly how the FTE is defined in the abstract? The idea is not so complicated and can be explained without equations.

  *Thank you for the suggestion. We've added the definition to the abstract.*

- page 2, l.44f: you might also cite Kapp et al. (2018) "Spatial verification of high resolution ensemble precipitation forecasts using local wavelet spectra" , who also use wavelets but specifically only look at ensemble predictions. They also project the fields to a lower-dimensional space and then apply standard multi-variate verification techniques (similar to your approach), but the interpretation of their results is not so straightforward in terms of correlation lengths.

  *The suggested reference has been included and is briefly discussed.*

- page 4, l.93f: please explain the relationship between the correlation length and the shape of the histogram in a little more detail. I think this point is pretty central and not completely obvious.

  *We have expanded the discussion of how deficiencies in the spatial structure of the ensemble (too much or too little spatial variability) is linked to the histogram shapes (e.g., see sections 2 & 3.3.1).*

- page 4, l.97f: are you sure that cases where forecast and observations never exceed the threshold should just be discarded entirely? Shouldn't the addition of

a few correct negative cases to the verification data-set lead to a better score for the prediction system?

*This depends on the purpose of the metric. If the estimated beta-score of an FTE histogram were to be interpreted as a score in the traditional sense, then we agree that one should give the ensemble system credit for correctly predicting no exceedances – in which case artificial flattening would actually be desirable. However, we don't see the FTE method as a quantitative metric; perhaps the CRPS of the FTE is more in line with this goal (e.g., Scheuerer and Hamill, 2018, their Table 3). Instead, it is our view that the FTE histogram is best utilized as a diagnostic tool and argue that we should remove uninformative cases that flatten the histogram.*

- page 5, table 1: Consider adding a third column which states the interpretation in terms of the correlation length or the quality of the forecast. However, I'm not sure what exactly to write for ab (see main comment 3).

*Table 1 and the surrounding discussion regarding interpretation of the beta-score/bias in terms of misrepresentation of spatial variability by the ensemble forecast fields have been updated, and a column with interpretations of the different shapes has been added.*

- page 5, l.125: If I understand correctly, the first step in this equation is only correct if F is a strictly monotonic function. What if the distribution contains a point mass, say at zero as for precipitation? Does that affect the rest of your argumentation? More generally, can you comment on how the results of your simulation study transfer to discontinuous variables like precipitation (as studied in section 4)?

*Our goal here is mainly to illustrate that our Gaussian process model is sufficiently general to simulate different weather variables, even if these are spatially inhomogeneous. We believe it is sufficiently general even for variables with a point mass (like precipitation). While our argument starts with the weather variable*

*of interest and shows how it can be transformed to standard Gaussian marginal distributions, the simulation actually works the other way round: we can simulate from a Gaussian process and transform the marginal distributions to those of the desired weather variable. This is always possible, even if there is a point mass; in this case values below a certain threshold simply get mapped to zero. We have added this clarification to the manuscript.*

- page 5, l.132: Do forecast and verification really need to be correlated for this experiment? What would happen if they weren't?

  *Please refer to the newly added Appendix D in which we examine a simulation experiment where the forecast and verification are explicitly uncorrelated. The short answer is that correlation is not necessary; the FTE is sensitive to miscalibration of the spatial structure, regardless of the value of correlation parameter (skill).*

- page 6, l.139: You should make it clear whether s and h are vectors or scalars, a process where C depends on a vector valued h would be anisotropic.

  *Thank you for pointing this out, our notation was not completely clean here. We fixed the problem.*

- page 6, l.147f: I don't understand the grammar in the sentence starting with "The multivariate ..." , is "Matern" the subject and "sets" the predicate?

  *Correct. We've updated the language for clarity.*

- page 7, l.184: Please clarify in what sense these fields are "realistic" .

  *The subjective term "realistic" has been removed in favor of a direct comparison to real data fields.*

- page 8, Fig.2: Please make it unambiguous in the figure or caption which parameter value is a and which is b.

*We've added the distinction to the caption.*

- page 9, l.205: You should discuss the origin and interpretation of this skewness (see general comment 3). It looks like forecasts whose only shortcoming lies in the correlation length obtain a left or right skew at certain thresholds (Fig. 5). What does that mean for the interpretation of a>b and a<b in realistic situations?

  *A more rigorous discussion on the origin and interpretation of skewness has been added, and Table 1 has been updated accordingly.*

- page 10, Fig.5: What's a0 here?

  *a0 = 2; we've added this in the caption.*

- page 11, l220: At least for the two most extreme ratios, a and b not only tend to one but go beyond and indicate the opposite bias. How do you interpret that? Can you maybe add error bars to these plots (via bootstrapping?) to check whether some of these effects might be insignificant?

  *Inclusion of the 95% confidence interval suggests that this tendency toward the opposite scoring is significant. We believe that this is due to the confounding effect of resolving ties at random, which happens more and more often (and between more and more tied FTE values) as the threshold increases, as only few ensemble member forecasts ever cross that threshold. This results in a massive distortion of histogram shapes, which still indicates non-uniformity, but makes it almost impossible to diagnose the source of this non-uniformity. We've added a discussion of this observation.*

- page 11, l221: While it is true that longer correlation lengths on the same grid lead to the same patterns as a decrease in domain size, are you sure that the number of available grid-points has no effect at all? Surely for very small domains it becomes increasingly hard to confidently estimate the spatial correlation structure?

*The number of available grid points is constant in our setup, what changes (as the correlation length of the verification field is varied) along with the 'effective domain size' is the spatial resolution. While we agree that this would have an effect on the ability to estimate the parameters of the underlying spatial correlation function, we note that such model identification is not the goal (and not required) here. For the FTE calculations, we think that the spatial resolution is not important, and the number of available grid points is only important to the extent that an insufficient number of grid points within the domain would increase the sampling variability in the calculation of the FTE values (and everything derived from them).*

- page 13, l272f: why not quantify the shape of the univariate histograms via a beta distribution as well?

  *The beta-characterization is now included with the histograms.*

- page 14, l294f: Can you discuss the skewness seen in Fig.9? The difference between a and b increases with threshold but, if I understand correctly, has a different sign than seen in Fig.5 (a>b but both are smaller than 1).

  *Skewness in Fig. 9 is observed primarily in months where the marginal calibration is already unreliable (i.e., Jan, April, Oct) which obfuscates interpretation of the skewness. As (now) noted in Table 1, skewness occurs at higher thresholds in combination with a U-shape, but it also occurs as a result of an under-forecast bias, and the univariate rank histograms suggest such an under-forecast bias for Jan, April, and Oct, and to a lesser degree also for July. We think that this effect is dominant here, and counteracts the spatial miscalibration-driven beta-bias effects that we would expect in conjunction with a U-shape. We have added some of this discussion to the paper.*

- page 14, l297f: I would also note that this kind of consideration could generally help you distinguish the effects of miscalibrated margins from errors in the spatial structure: If the miscalibration seen in figure 9 were caused by errors in the

marginal distributions, one might expect the effect to be weakest in July were the marginal calibration was the best.

*Correct – summer is really the only season in this example that permits an un-ambiguous conclusion about the spatial structure. This important point is given additional emphasis in our discussion (see final lines of section 4.2).*

- page 15, Fig.9: Some of these histograms are clearly not uni-modal. Can you comment on how your method is affected by cases where the beta-fit is likely not very good?

*We believe this is just a result of sampling variability. If there is truly a multimodal shape of the histogram that is not well-represented by a beta distribution, it won't be captured by the beta-fit / beta-characterization. However, this could also be an advantage if this is merely due to sampling variability; in this case the beta distribution fit filters out some of that variability.*

**Reviewer 2**

This manuscript discusses multivariate verification of ensemble forecasts with a focus on spatial structures. A new approach is proposed, illustrated and discussed using synthetic and precipitation datasets.

The proposed approach consists in transforming a multivariate quantity (a spatial field) into a univariate quantity. This is performed by thresholding and counting the fraction of grid points exceeding a threshold over a domain. The popular rank histogram tool is applied to the resulting ensemble and observed fractions of threshold exceedance (FTE). The method is easy to implement and is appealing because of the simplicity of both the interpretation of the derived univariate product and of the derived rank

histograms.

In order to better understand FTE histograms, a toy model is used to assess the sensitivity of the method to predefined discrepancies in a controlled environment. In a real-life example context, FTE histograms are derived for precipitation forecasts and limitations of the underlying forecasting system diagnosed. The paper is well-written, well-structured, and pleasant to read. The description of the method and datasets is clear, the discussion of the results convincing.

However, the manuscript would benefit from some clarifications. First, the authors could clarify the link between the FTE histogram approach and the fraction skill score (FSS, Roberts and Lean 2008), a popular verification metric applied to deterministic forecast of precipitation fields. The first step is similar for the two methods (transformation of a real-value field into a binary field and then a focus on the fraction of events over a domain). It would be beneficial for the verification community to clearly state the link between FTE histogram and FSS.

*Thank you for highlighting this connection; a brief discussion has been added to the introduction.*

Secondly (and more importantly), the author could discuss the impact of miscalibration of the univariate distribution on the interpretation of the FTE. It is clearly mentioned in the text that this aspect should not be disregarded. In line 86, it is stated that the FTE histogram can be used "once the marginal distribution has been checked" , and in Section 4.2 that "the possible non-uniformity of FTE histograms [...] could be due to univariate miscalibration". Perfect calibration of univariate distributions cannot be
expected in reality so a discussion about the permeability of FTE histograms to univariate miscalibration would be more than useful for the interpretation of real case FTE histograms.

*We have expanded the discussion of miscalibration of the univariate forecast distributions and its effect on the shape of the FTE histograms (both in the main text and in Table 1). When both univariate distributions and spatial correlations are miscalibrated, it is difficult to disentangle these effects. This is a consequence of projecting a multivariate quantity onto a univariate one which inevitably results in some loss of information. This is why we strongly advocate for considering different verification metrics - including univariate rank histograms - alongside each other. This prevents false conclusions in the worst case scenario where marginal and spatial miscalibration cancel each other out. We note, however, that even in this case the FTE histograms (unlike other multivariate verification metrics) permit conclusions about the representation of forecast uncertainty about quantities that are meaningful in practice, so while it is not always straightforward to diagnose the exact source of mis-calibration, one can still use FTE histograms in conjunction with metrics that study e.g. regional averages or maxima to compare and rank the quality of different forecast systems.*

In addition, the authors could clarify why the parameters derived from the beta distribution fitting differ from the parameters defined in the Keller and Hense 2011 paper. The list of references could also be checked and the duplicated doi information removed.

*We've made an overarching transition throughout our discussion and analysis to the beta-score and beta-bias defined in Keller and Hense*

Interactive
comment
*(2011). The duplicated doi information has also been removed from the list of references.*

Please also note the supplement to this comment:
https://npg.copernicus.org/preprints/npg-2019-63/npg-2019-63-AC1-supplement.pdf

**Supplement:**

[revised manuscript text omitted]